# Substrate-binding destabilizes the hydrophobic cluster to relieve the autoinhibition of bacterial ubiquitin ligase IpaH9.8

Yuxin Ye [1,2,3,4✉], Yuxian Xiong[1,3,4] & Hao Huang [1,3✉]

IpaH enzymes are bacterial E3 ligases targeting host proteins for ubiquitylation. Two auto-inhibition modes of IpaH enzymes have been proposed based on the relative positioning of the Leucine-rich repeat domain (LRR) with respect to the NEL domain. In mode 1, substrate-binding competitively displaces the interactions between theLRR and NEL to relieve auto-inhibition. However, the molecular basis for mode 2 is unclear. Here, we present the crystal structures of *Shigella* IpaH9.8 and the LRR of IpaH9.8 in complex with the substrate of human guanylate-binding protein 1 (hGBP1). A hydrophobic cluster in the C-terminus of IpaH9.8[LRR] forms a hydrophobic pocket involved in binding the NEL domain, and the binding is important for IpaH9.8 autoinhibition. Substrate-binding destabilizes the hydrophobic cluster by inducing conformational changes of IpaH9.8[LRR]. Arg166 and Phe187 in IpaH9.8[LRR] function as sensors for substrate-binding. Collectively, our findings provide insights into the molecular mechanisms for the actication of IpaH9.8 in autoinhibition mode 2.

[1] State Key Laboratory of Chemical Oncogenomics, School of Chemical Biology and Biotechnology, Peking University Shenzhen Graduate School, 518055 Shenzhen, China. [2] Shenzhen Bay Laboratory Pingshan Translational Medicine Center, Shenzhen, China. [3] Laboratory of Structural Biology and Drug Discovery, Peking University Shenzhen Graduate School, 518055 Shenzhen, China. [4]These authors contributed equally: Yuxin Ye, Yuxian Xiong. ✉email: yeyx@pkusz.edu.cn; huang.hao@pku.edu.cn

Ubiquitination, as one of the most important protein post-translational modifications, regulates a multitude of biological processes in eukaryotic cells[1–3]. In the ubiquitin-proteasome system (UPS), the 76-amino acid ubiquitin (Ub) is conjugated to lysine residues in target proteins. Ubiquitination is mediated by a three-enzyme cascade consisting of the Ub-activating enzyme (E1), the Ub-conjugating enzyme (E2), and the Ub-protein ligase (E3)[4–6]. E3s play a crucial role in this cascade for their precise control of both the efficiency and substrate specificity in ubiquitination[7]. E3s can be grouped into three classes: the RING (Really Interesting New Gene) E3s, the HECT (Homologous to E6AP C-terminus) E3s, and the RBR (Ring-Between-Ring) E3s[8]. These three classes of E3s use two distinct strategies to transfer Ub. The RING E3s, which are characterized by their RING or U-box catalytic domain, directly transfer Ub from an E2 to the target substrate[9–11]. In contrast, the HECT and RBR E3s, both of which utilize active site cysteines, catalyze ubiquitination in a two-step strategy: (1) transfer Ub from the E2–Ub to the E3 catalytic cysteine residue in a transthiolation reaction and (2) then transfer Ub from the E3–Ub to the substrate lysine (or the N terminus) in a subsequent aminolysis reaction[12–14].

Ubiquitination plays important roles in pathogenic infection[15]. Host cells can utilize the UPS to activate innate immune responses against bacterial invasion, but invasive pathogens can also use the UPS against the host in order to facilitate infection[16–19]. The IpaH family is a group of conserved effectors that are secreted by Shigella, Salmonella, and other Gram-negative bacteria through type III secretion systems[20,21]. IpaH proteins function as E3 ligases and subvert the host UPS pathway to suppress the host inflammasome and innate immune responses[22]. These enzymes contain an N-terminal type III secretion signal sequence, a leucine-rich repeats (LRR) domain involved in substrate recognition, and a C-terminal conserved novel E3 ligase (NEL) domain[23]. Although the NEL domain has little sequence or structural homology to known HECT E3s, IpaH enzymes possess E3 ligase activity that is dependent on a catalytic cysteine in the NEL domain, which facilitates Ub transfer from E2 to the substrate in a manner similar to HECT E3s[23–25].

Autoinhibition is an important regulatory mechanism by which IpaH enzymes prevent self-degradation from the host UPS or activation of the host innate immune response via formation of free polyUb chains[26,27]. Two distinct autoinhibition modes have been proposed based on the crystal structures of SspH2 (mode 1) and IpaH3 (mode 2)[28,29]. In SspH2 structure (mode 1), the concave surface of the LRR domain orients toward the NEL domain, whereas in IpaH3 structure (mode 2), the concave surface is oriented in an opposite direction relative to the NEL domain[27]. Binding of the substrate to the LRR domain could activate the IpaH in both autoinhibition modes[27,30]. Previously, the crystal structure of SspH1^LRR-PKN1^HR1b revealed that substrate-binding competitively displaces interactions between the LRR and NEL domains, thereby relieving autoinhibition in mode 1 (ref. [31]). However, the mechanistic details that explain how substrate binding triggers IpaH activation in mode 2 are missing.

IpaH9.8 is one of the most well-studied members of the IpaH family and is secreted by Shigella flexneri during infection[23]. IpaH9.8 targets Ste7 for degradation in order to hijack the host UPS and subsequently inhibit mitogen-activated protein kinase (MAPK)-dependent signaling pathways in yeast[23]. Additional studies have shown that IpaH9.8 plays an important role in modulating host inflammatory responses by ubiquitinating NF-κB essential modulator (NEMO) and then promoting NEMO degradation[32]. Recently, it has been reported that IpaH9.8 targets human guanylate-binding protein 1 (hGBP1) for ubiquitination and degradation in order to suppress host defense in cells[33–35]. As the best-characterized member of the GBP family, hGBP1 is a GTPase

with both antiviral and antibacterial activities that is encoded by an interferon-activated human gene[36]. hGBP1 consists of a large GTPase domain (LG) at the N-terminus, an α-helical middle domain (including α7–α11, MD), and an α-helical GTPase effector domain (including α12–α13, GED) at the C-terminus[37–39]. As one member of the dynamin superfamily, hGBP1 greatly increases GTP hydrolysis activity upon oligomerization[38,40,41]. When hGBP1 binds to different guanine nucleotides or transition state analogs, it adopts different oligomer/monomer states[37,42–45]. Previous biochemical work has revealed that the recognition of hGBP1 by IpaH9.8 is independent of the nucleotide-bound state of hGBP1 (ref. [34]). However, there is a dearth of direct structural evidence supporting this concept.

IpaH9.8 is 82% identical to IpaH3, suggesting that the IpaH9.8 might autoinhibit its enzymatic activity similarly to IpaH3 in mode 2 (ref. [46]). In this study, we solved the crystal structure of the near full length IpaH9.8, and observed that the architecture of IpaH9.8 is similar to IpaH3 in autoinhibition mode 2. Based on the structure of IpaH9.8, we investigated the importance of a hydrophobic cluster in the C-terminal region of IpaH9.8^LRR for IpaH9.8 autoinhibition. Then, to study the activation mechanism of IpaH9.8 in mode 2, we solved the crystal structure of the LRR domain of IpaH9.8 (IpaH9.8^LRR) in complex with full-length hGBP1. By comparing the structures of IpaH9.8^LRR in apo and substrate-bound forms, we illuminated the mechanisms by which substrate activates IpaH9.8 via induction of conformational changes in IpaH9.8^LRR. Further structural analysis revealed that two residues located in the concave surface of IpaH9.8^LRR function as sensors for substrate binding. Finally, we described the molecular basis for Ub transfer from NEL to substrate after activation. Collectively, this study reveals a substrate-induced activation mechanism of IpaH9.8 in mode 2.

## Results
**Overall structure of IpaH9.8.** The near full-length structure of IpaH9.8 (truncated the N-terminal T3SS signal sequence) was determined by crystallization and X-ray diffraction at a 2.75 Å resolution (Fig. 1A). The crystals belonged to a space group I222 with one copy of IpaH9.8 in each asymmetric unit. The final refined model of IpaH9.8 consisted of Thr22–Asp244 within the LRR domain, and Ala255–Ser536 within the NEL domain. The LRR domain of IpaH9.8 adopted an expected solenoid-like structure consisting of eight LRR motifs, with both the N-terminus and C-terminus capped by two helices (Fig. 1A, right). For the convenience of later sections, the N-terminal cap region with LRR1–LRR6 is defined as LRR-NT, with the C-terminal cap region with LRR7–LRR8 defined as LRR-CT. The NEL domain of IpaH9.8 is an all helical-fold structure, consisting of three subdomains: N-subdomain (from α5 to α8), M-subdomain (from α9 to α13), and C-subdomain (from α14 to α17) (Fig. 1A, left)[30]. In a previous study, the isolated NEL domain of IpaH9.8 (IpaH9.8^NEL-iso) was crystallized in non-reducing conditions and reported to be an inactive, likely an artificial structure[47]. In the IpaH9.8^NEL-iso structure, the active cysteine from two different asymmetric units formed a disulfide bond and caused α8 and α9 to extend into a long straight helix (Fig. 1B). However, our Native-PAGE results confirmed the presence of an IpaH9.8 (and IpaH9.8-C337A) oligomer under both reducing and non-reducing conditions (Supplementary Fig. 1A, B), suggesting that the IpaH9.8 oligomer is irrelevant to intermolecular disulfide bonds. In our structure, the α8 and α9 in the NEL domain of the IpaH9.8 (termed IpaH9.8^NEL-FL) showed a "helix-loop-helix" fold, with the loop region between α8 and α9 being flexible and not observable in electron density map (Fig. 1A, right; 1B). Compared to the structure of IpaH9.8^NEL-FL, the N-subdomain of IpaH9.8^NEL-iso rotated upward about 130°, which

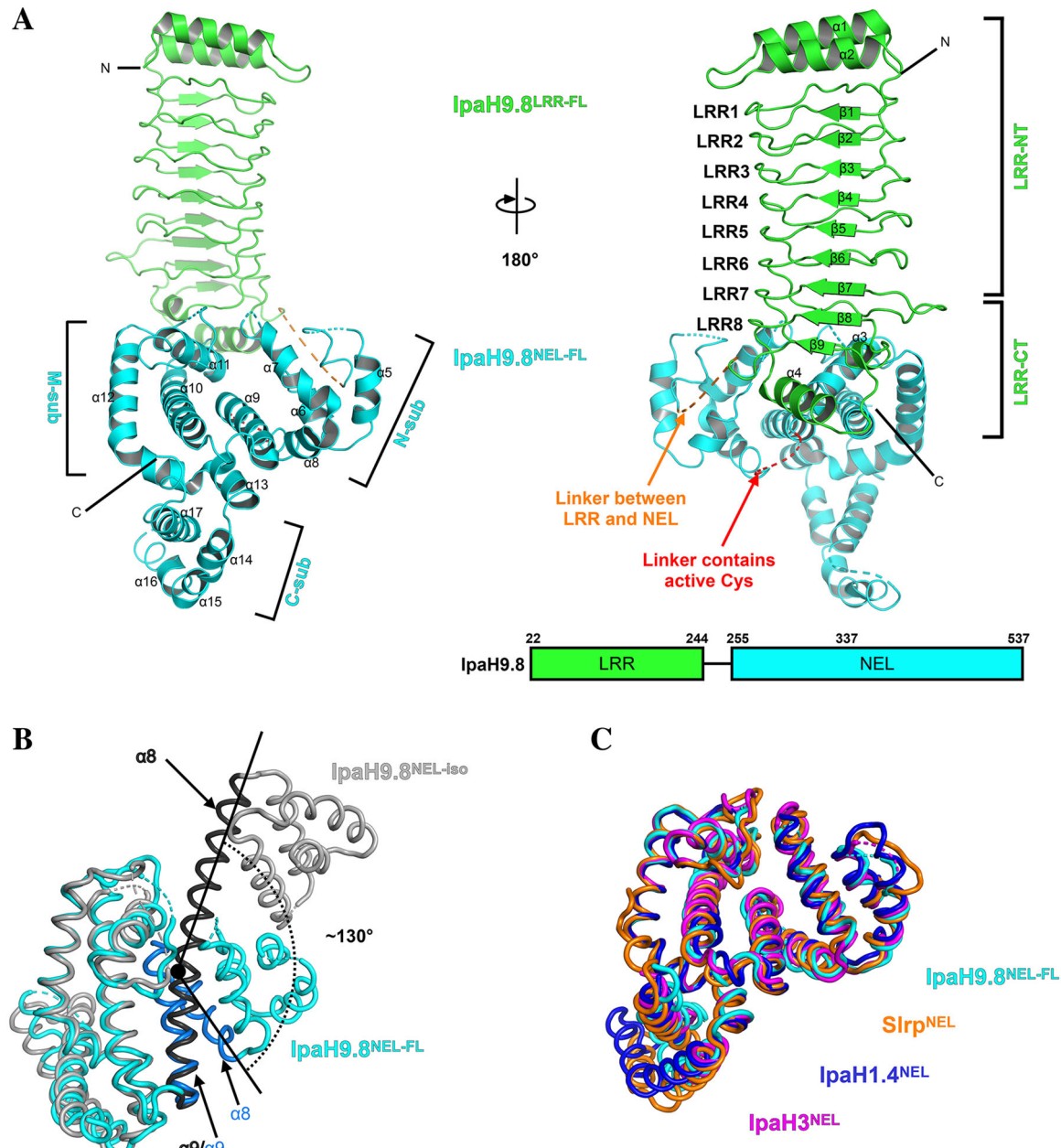

**Fig. 1 Overall structure of IpaH9.8. A** Top, schematic representations of the IpaH9.8 constructs used for crystallization. The LRR domain is colored green and the NEL domain cyan. Left, a cartoon model of near full-length IpaH9.8 with domains colored as described above. Three NEL subdomains (N-sub, M-sub, and C-sub) and the α-helixes (α5–α17) in NEL are indicated. Right, cartoon representation of IpaH9.8 as in left but rotated ~180° with the LRR motifs, LRR-NT, and LRR-CT labeled. The α-helixes (α1–α4) and the β-sheets (β1–β9) in LRR are indicated. Linkers without a determined structure are indicated by dashed lines with the linker between LRR and NEL colored orange, while the linker containing the active Cys residue is colored red. **B** Superposition of IpaH9.8[NEL-FL] (cyan) and IpaH9.8[NEL-iso] (PDBID: 3L3P, gray) structures. The helixes α8/α9 in IpaH9.8[NEL-FL] and IpaH9.8[NEL-iso] are labeled and colored blue and black, respectively. The α8/α9 bends ~130° in IpaH9.8[NEL-FL] but adopts a straight α-helical conformation in IpaH9.8[NEL-iso]. **C** Superposition of IpaH9.8[NEL-FL](cyan), Slrp[NEL] (PDBID: 4PUF, orange), IpaH1.4[NEL] (PDBID: 3CKD, blue) and IpaH3[NEL] (PDBID: 3CVR, magenta) structures.

results in a clash with the LRR domain (Fig. 1B and Supplementary Fig. 1D). Moreover, the high similarity in structure of the NEL domain of IpaH proteins (Fig. 1C) is consistent with the sequence conservation observed in the IpaH family (Supplementary Fig. 2). Collectively, these results suggested that the NEL domain of IpaH9.8[NEL-FL] is more likely to be in a native state.

**The hydrophobic cluster in LRR-CT is important for IpaH9.8 autoinhibition.** The autoinhibition mode of IpaH9.8[NEL-FL] is similar to IpaH3 in mode 2 (Supplementary Fig. 3A). Structural analysis revealed that Phe395 in the NEL domain makes hydrophobic contact with Ile196 in LRR-CT (Fig. 2A), which might be important for IpaH9.8 autoinhibition. Indeed, single amino acid substitutions of Ile196 (I196D) and Phe395 (F395R) weakened IpaH9.8 autoinhibition, which may be caused by disrupting the intermolecular interaction between LRR and NEL (Fig. 2B)[27]. However, the mutants I196A and F395A did not affect the autoinhibition (Fig. 2B). Further structural analysis revealed that six hydrophobic residues (Ile196, Leu201, Ile211, Leu216, Leu224,

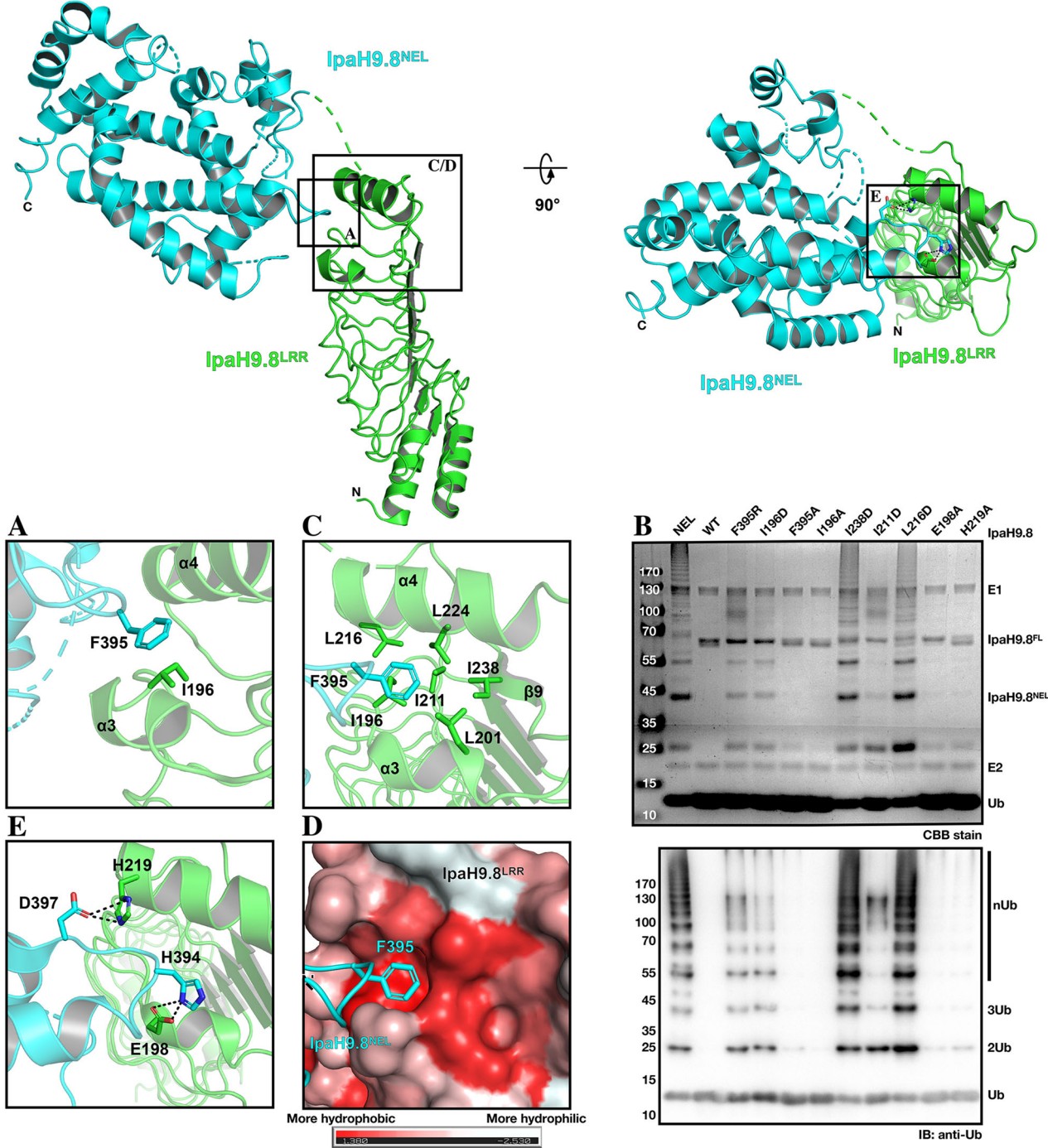

**Fig. 2 The hydrophobic cluster in the LRR-CT is important for autoinhibition of IpaH9.8. A** Close-up view of the LRR–NEL interface of IpaH9.8, with Ile196 in LRR and Phe395 in NEL represented as sticks and colored green and cyan, respectively. The α3 and α4 in LRR-CT are labeled. **B** In vitro ubiquitination assays with the full-length IpaH9.8 or the NEL domain of IpaH9.8. Reaction products were detected by Coomassie Brilliant Blue (CBB) staining (top) and blotted with anti-ubiquitin antibody (bottom). **C** Magnified view of the LRR-CT of IpaH9.8 with α3, α4, and β9 labeled. The hydrophobic cluster residues in LRR are represented as sticks and colored in green. Phe395 in NEL is colored in cyan. **D** The same view of IpaH9.8 as in **C** but the LRR surface is colored based on hydrophobicity (1.38 to −2.53; red to white, red indicating hydrophobicity and white indicating hydrophilicity). **E** Magnified view of LRR-CT. The residues in LRR are represented as sticks and colored green, and the residues in NEL are colored cyan. Dashed lines indicate hydrogen bonds.

and Ile238) located in LRR-CT constitute a hydrophobic cluster (Fig. 2C). In addition, this hydrophobic cluster forms a hydrophobic pocket in which the Phe395 of NEL could be confined (Fig. 2D). We speculate that the substitution of Ile196 or Phe395 with another hydrophobic residue alanine (I196A, F395A) might not destabilize the hydrophobic cluster nor disrupt the interaction

between LRR and NEL, whereas substitution with hydrophilic residues (I196D, F395R) may destabilize the hydrophobic cluster and disrupt the LRR and NEL interaction. The disruption by I238D has been explained as a part of mode 1 autoinhibition, as observed by the importance of Ile479 in SspH2 (corresponding to Ile238 of IpaH9.8) for interacting with Leu638 of SspH2^NEL in

mode 1 (Supplementary Fig. 3B)[27]. However, given the hydrophobic cluster mentioned above, the abrogation of autoinhibition of IpaH9.8 by mutation I238D is more likely due to the destabilization of the hydrophobic cluster.

In order to investigate the importance of the hydrophobic cluster in IpaH9.8 autoinhibition, two hydrophobic residues, Ile211 and Leu216, which contribute to the hydrophobic cluster formation but is distal from the LRR–NEL interface in both IpaH9.8 and SspH2 were mutated to aspartate residues, i.e. I211D and L216D (Fig. 2C and Supplementary Fig. 3C). In vitro polyubiquitin chain synthesis assays revealed that I211D moderately weakened the autoinhibition of IpaH9.8, while the autoinhibition in L216D was vastly released (Fig. 2B). This suggested that the hydrophobic cluster was important for IpaH9.8 autoinhibition.

In addition to the hydrophobic interactions, analysis of the IpaH9.8 structure also reveals that Glu198 and His219 form hydrogen bonds on the LRR–NEL interface with His394 and Asp397, respectively (Fig. 2E). Specifically, a previous report revealed that Asp397 functions as a catalytic base in IpaH9.8 aminolysis, which is a limiting step of autoinhibition[48]. Therefore, the interactions introduced by His394 or Asp397 may benefit autoinhibition[48]. Unexpectedly, polyUb chain test showed that the mutants E198A and H219A have no observable effects on IpaH9.8 autoinhibition (Fig. 2B), indicating that the autoinhibition is governed by the LRR–NEL conformation, which is dominated by hydrophobic contacts instead of hydrogen-bond interactions. This result further highlighted the importance of the hydrophobic cluster for IpaH9.8 autoinhibition.

**Overall structure of IpaH9.8^LRR–hGBP1 complex**. To obtain the complex structure of IpaH9.8–substrate, we initially attempted to crystalize the full-length (or near full-length) IpaH9.8 in complex with the full-length hGBP1 but did not succeed. Crystals were obtained for the complex of the IpaH9.8 LRR domain and the full-length hGBP1, and the structure of the IpaH9.8^LRR–hGBP1 complex was determined at 3.72 Å (Fig. 3A). The complex crystals belong to a space group $P6_122$ and contained one heterodimer in each asymmetric unit. Coincidentally, one month after this complex structure of IpaH9.8^LRR–hGBP1 was solved, a structure of IpaH9.8^LRR in complex with the LG-MD region of hGBP1 (IpaH9.8^LRR–hGBP1^LG-MD) at 3.6 Å was reported[49]. This allowed for a more systematic structural analysis of the IpaH9.8^LRR–substrate complex under different conditions.

The main difference between the IpaH9.8^LRR–hGBP1 and the IpaH9.8^LRR–hGBP1^LG-MD complex structures is within the MD domain of hGBP1 (Supplementary Fig. 4A). Specifically, the α7 of the hGBP1-MD in IpaH9.8^LRR–hGBP1 and IpaH9.8^LRR–hGBP1^LG-MD is rotated ~10° and ~30° with respect to hGBP1-apo, respectively, which results in the entire MD domain bending toward the LG domain to different extents (Fig. 3B, left). In addition, the α12 of the hGBP1-GED domain in IpaH9.8^LRR–hGBP1 rotated ~10° with respect to hGBP1-apo (Fig. 3B, right). The MD domain in IpaH9.8^LRR–hGBP1^LG-MD bends more than in IpaH9.8^LRR–hGBP1, possibly because the GED domain restricts the flexibility of the MD domain. Previous reports have noted that the flexibility of the GED and MD domains is essential to the hGBP1 oligomerization[37]. Together, these findings are consistent with the GED and MD domains being highly plastic.

The structural comparison reveals that the two complex structures correspond to one another in how IpaH9.8^LRR–hGBP1 binds at the interface (Supplementary Fig. 4B). Similar to the IpaH9.8^LRR–hGBP1^LG-MD complex, the LG domain of hGBP1 in the IpaH9.8^LRR–hGBP1 complex interacted with the concave side of the IpaH9.8^LRR mainly through the P-loop, the Switch II, and

the α3 helix (Supplementary Fig. 4C). However, the IpaH9.8^LRR–hGBP1^LG-MD structure lacks a GDP molecule that was observed at the catalytic center of the LG domain in the IpaH9.8^LRR–hGBP1 complex (Fig. 3A, C and Supplementary Fig. 4D). Further structural analysis revealed that there were different conformations between the two complex structures for the guanine-cap and phosphate-cap (often referred to as Switch I) in the LG domain, which are important for nucleotide binding, dimerization, and self-activation of the GTPase activity for hGBP1 (ref. [43]). In the IpaH9.8^LRR–hGBP1 complex, the guanine-cap was visible and revealed a similar orientation to other nucleotide-bound structures of hGBP1 (Fig. 3C, D and Supplementary Fig. 4E), and the phosphate-cap was also visible but flipped compared with other nucleotide-bound hGBP1 structures (Fig. 3C, E and Supplementary Fig. 4F). This difference may have been caused by the lack of phosphate, which would normally bind to the phosphate cap. In contrast, in the IpaH9.8^LRR–hGBP1^LG-MD complex, neither cap region was visible as in the hGBP1-apo structure (Fig. 3C–E). In addition, both complex structures revealed that binding IpaH9.8 could occupy the dimer interface of hGBP1 (ref. [49]). These observations provide direct structural evidence that the recognition of hGBP1 by IpaH9.8 is independent of its nucleotide-bound state[34].

**Substrate-binding induced the destabilization of the hydrophobic cluster in LRR-CT**. In order to investigate the activation mechanism of IpaH9.8, the complex structure was compared across multiple states: in IpaH9.8^LRR–hGBP1, named IpaH9.8^LRR-Sub; in IpaH9.8^LRR–hGBP1^LG-MD, named IpaH9.8^LRR-Sub'; in IpaH9.8^FL, named IpaH9.8^LRR-FL; and in isolated IpaH9.8^LRR (ref. [50]), named IpaH9.8^LRR-iso. Comparison of these LRR domains unmasked great conformational differences in the LRR-CT during substrate engagement (Fig. 4A, left). Specifically, the LRR-CT in substrate-binding forms (IpaH9.8^LRR-Sub and IpaH9.8^LRR-Sub') rotated ~10 Å and ~20 Å and translates ~5 Å and ~10 Å toward the convex side with respect to IpaH9.8^LRR-FL and IpaH9.8^LRR-iso, respectively (Fig. 4A, right). As the hydrophobic cluster in the LRR-CT is important for IpaH9.8 autoinhibition, conformational changes induced by substrate binding might affect the stability of the hydrophobic cluster and LRR-CT. The normalized B-factors of Cα atoms revealed that the LRR-CTs of IpaH9.8^LRR-Sub and IpaH9.8^LRR-Sub' were more flexible than the other two LRR domains (Fig. 4B).

To probe the effect caused by substrate binding to the hydrophobic cluster, a detailed structural analysis was performed exploring interactions at the hydrophobic cluster. As the side-chain density map of the hydrophobic cluster residues in IpaH9.8^LRR-Sub was not clear, only the structure of IpaH9.8^LRR-Sub' was used to analyze the hydrophobic cluster (Supplementary Fig. 5A). Superimposition of LRR-CTs from IpaH9.8^LRR-Sub', IpaH9.8^LRR-FL, and IpaH9.8^LRR-iso revealed that the hydrophobic cluster residue side chains in IpaH9.8^LRR-Sub' were shifted slightly with respect to IpaH9.8^LRR-FL and IpaH9.8^LRR-iso (Fig. 5A). These seemingly subtle movements resulted in clear differences in the cavities created by the hydrophobic cluster. Specifically, the cavities in IpaH9.8^LRR-Sub' were more in number and less compact in size than those in IpaH9.8^LRR-FL and IpaH9.8^LRR-iso, respectively (Fig. 5B). More hydrophobic cavities were generally accompanied with more buried waters and high free energy[51–53], suggesting that the hydrophobic cluster in IpaH9.8^LRR-Sub' was less stable than those of IpaH9.8^LRR-FL and IpaH9.8^LRR-iso.

Next, the destabilization of the LRR-CT, especially within the hydrophobic cluster, was investigated for its effect on the LRR–NEL interface. As mentioned previously, the hydrophobic pocket in LRR-CT serves as a binding pocket for the Phe395 in the

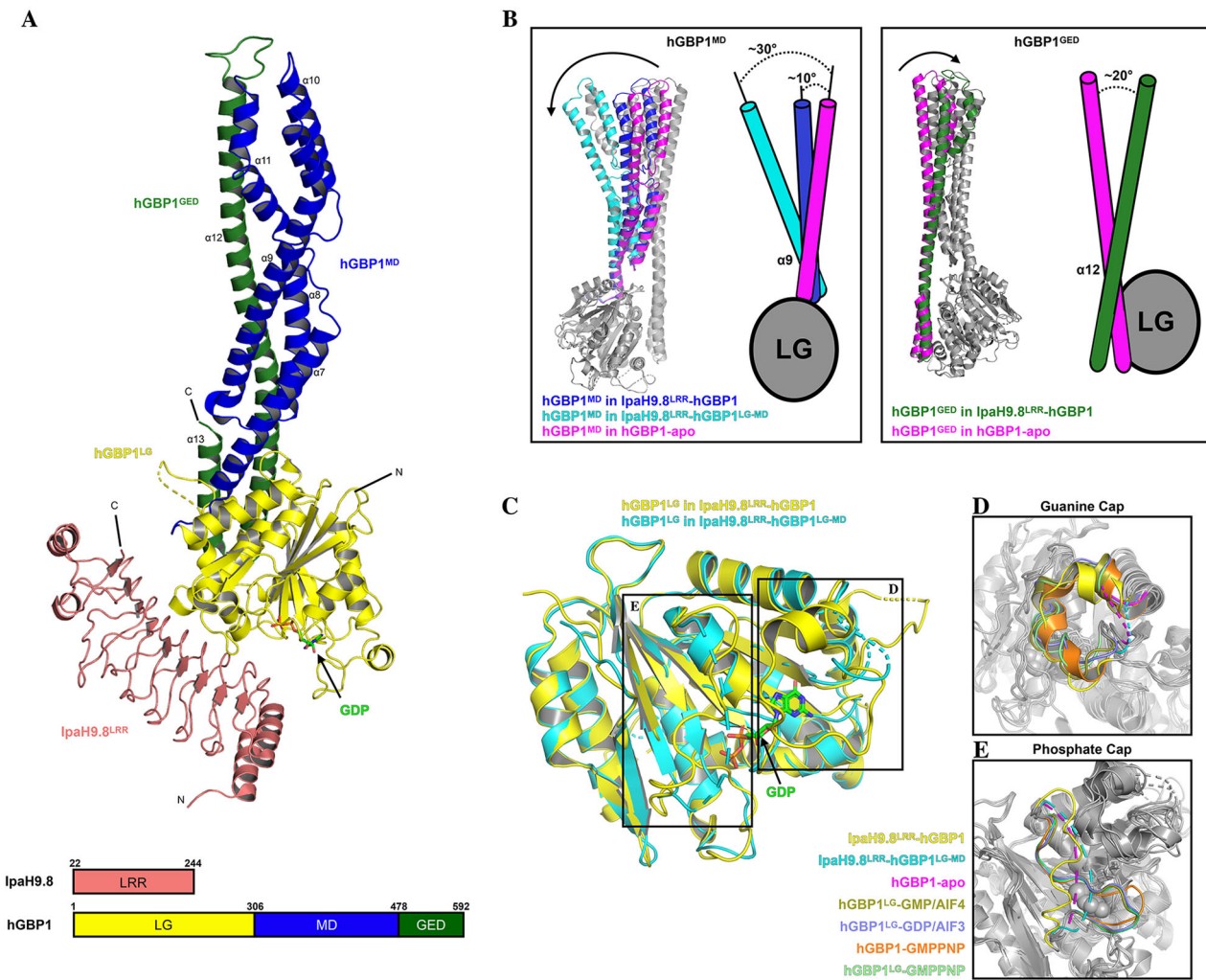

**Fig. 3 Overall structure of IpaH9.8$^{LRR}$–hGBP1 complex. A** Top, the cartoon model of IpaH9.8$^{LRR}$ and hGBP1 complex, with IpaH9.8$^{LRR}$ colored salmon, hGBP1$^{LG}$ yellow, hGBP1$^{MD}$ blue, and hGBP1$^{GED}$ forest. The GDP molecular structure is represented as sticks (C: green, O: red, P: orange, N: blue). The α-helixes (α7–α13) in hGBP1$^{MD}$ and hGBP1$^{GED}$ are labeled as such. Bottom, schematic representation of the IpaH9.8$^{LRR}$ and hGBP1 constructs for crystallization. **B** Left frame—left, superimposition of the structures of hGBP1 by the LG domain, with the MDs differentiated by colors. Arrows indicate the rotation of the MDs when bound to IpaH9.8$^{LRR}$. Left frame—right, cartoon schematic of the difference in angles of α9 for each hGBP1, as in left frame-left. Right frame—left, superimposition of the structures of hGBP1 by the LG domain with the GED domains differentiated by colors. Arrows indicate the rotation of GED when bound to IpaH9.8$^{LRR}$. Right frame—right, cartoon schematic of differences in angle of α12 for each hGBP1, as in right frame-left. **C** Superimposition of hGBP1$^{LG}$ with IpaH9.8$^{LRR}$–hGBP1 (yellow) and IpaH9.8$^{LRR}$–hGBP1$^{LG-MD}$ (cyan), with a GDP molecule in IpaH9.8$^{LRR}$–hGBP1 represented as sticks. **D** The guanine caps in different hGBP1$^{LG}$ domains are differentiated by colors. **E** The phosphate caps in different hGBP1$^{LG}$ domains are differentiated by colors. Structure sources: IpaH9.8$^{LRR}$–hGBP1 (this study), IpaH9.8$^{LRR}$–hGBP1$^{LG-MD}$ (PDBID: 6K2D), hGBP1-apo (PDBID: 1DG3), hGBP1–GMPPNP (PDBID: 1F5N), hGBP1$^{LG}$–GMPPNP (PDBID: 2BC9), hGBP1$^{LG}$–GDP/AlF3 (PDBID: 2B92), and hGBP1$^{LG}$–GMP/AlF4 (PDBID: 2B8W).

NEL domain to inhibit IpaH9.8. The LRR-CT of IpaH9.8$^{LRR-Sub}$ and IpaH9.8$^{LRR-iso}$ were superimposed onto the IpaH9.8$^{FL}$ structure (Fig. 5C), revealing that the hydrophobic pocket of IpaH9.8$^{LRR-Sub}$ was shallower than in IpaH9.8$^{LRR-FL}$ (Fig. 5D, left). Surprisingly, in the IpaH9.8$^{LRR-Sub}$ structure, this hydrophobic pocket demonstrated minor overlap with the Phe395 from the NEL domain (Fig. 5D, middle). Similar observations were also made for IpaH9.8$^{LRR-Sub'}$ (Supplementary Fig. 5B). In contrast, the pocket in IpaH9.8$^{LRR-iso}$ was quite similar in IpaH9.8$^{LRR-FL}$ and demonstrated no clash with the NEL domain (Fig. 5D, right), indicating that different conformations of the hydrophobic pocket were not due to differences in the NEL domain, but because of substrate-binding. Altogether, these results demonstrated that substrate-binding induced conformational changes in LRR-CT, which resulted in destabilization of the hydrophobic cluster and LRR-CT, which could disrupt the LRR–NEL interaction.

**Substrate-binding affected the stability of LRR-CT in IpaH9.8 by triggering conformational changes in the Arg166 and Phe187 of IpaH9.8$^{LRR}$.** Structural analysis of the hGBP1/IpaH9.8$^{LRR}$ interface in both complex structures of IpaH9.8$^{LRR}$–hGBP1 and IpaH9.8$^{LRR}$–hGBP1$^{LG-MD}$ revealed that hGBP1 mainly interacted with the LRR-NT of IpaH9.8, rather than LRR-CT (Supplementary Fig. 6A)[49]. To investigate the mechanism of substrate-binding affecting the stability of LRR-CT, analysis of Cα dihedral angle of the LRRs of different complexes revealed that three residues near the LRR-hGBP1 interface (Thr165, Arg166, and Asn167) were with different angles between IpaH9.8$^{LRR-FL}$ and IpaH9.8$^{LRR-Sub}$ (Fig. 6A, top). In contrast, the main chain structures of these residues were not different between IpaH9.8$^{LRR-FL}$ and IpaH9.8$^{LRR-iso}$ (Fig. 6A, bottom), which indicated that the different dihedral angles were caused by substrate-binding. Indeed, in the IpaH9.8$^{LRR}$–hGBP1 complex,

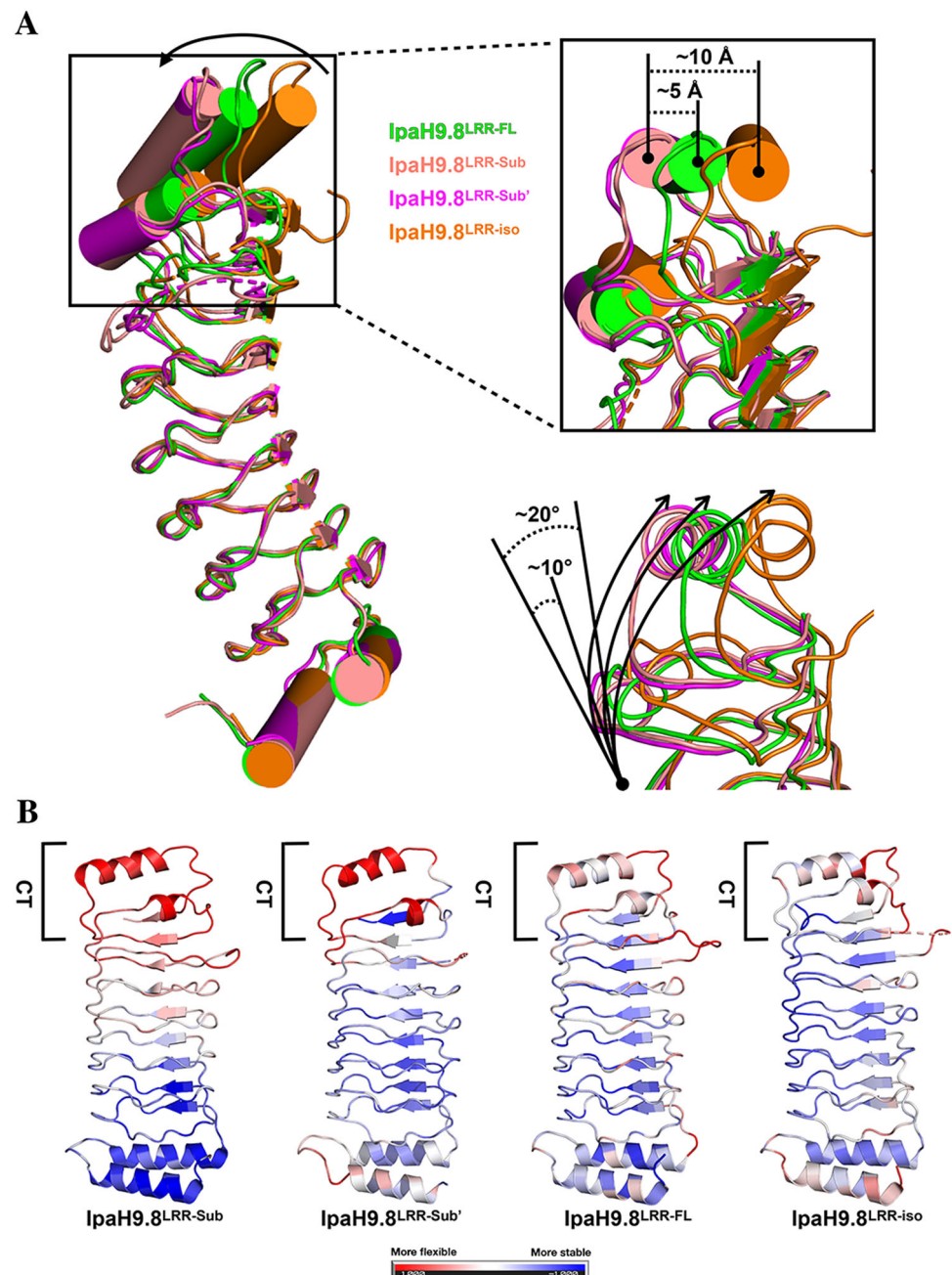

**Fig. 4 Substrate-binding destabilized the hydrophobic cluster in LRR-CT. A** Left, superimposition of the structures of IpaH9.8$^{LRR-Sub}$, IpaH9.8$^{LRR-Sub'}$ (PDBID: 6K2D), IpaH9.8$^{LRR-FL}$, and IpaH9.8$^{LRR-iso}$ (PDBID: 5B0T) by the LRR-NT with the LRR domains differentiated by colors. Arrows indicate the rotation of LRR-CT when bound to hGBP1. Right-top and right-bottom, magnified view of LRR-CTs in different IpaH9.8$^{LRR}$ constructs as in left. Relative differences in shift distance (top) and rotation angles (bottom) for LRR-CT are indicated. **B** The cartoon models of IpaH9.8$^{LRR-Sub}$, IpaH9.8$^{LRR-Sub'}$, IpaH9.8$^{LRR-FL}$, and IpaH9.8$^{LRR-iso}$ are colored by normalized B-factor (1 to 0 to −1; red to white to blue, red indicating flexibility and white indicating stability).

the Glu147 of hGBP1 formed hydrogen-bond interactions with the Arg166 of IpaH9.8$^{LRR}$ (Supplementary Fig. 6B, top), triggering main chain conformational changes in Arg166 and two other adjacent residues. A close inspection of the three noted residues revealed that the main chain carbonyl of Thr165 in LRR rotated ~180° and the main chain carbonyl of Arg166 in IpaH9.8$^{LRR-sub}$ shifted ~1 Å with respect to IpaH9.8$^{LRR-FL}$ and IpaH9.8$^{LRR-iso}$. These conformational changes broke three hydrogen-bonds: one between the main chain carbonyl of Asp189 and the main chain nitrogen of Arg166, one between the main chain carbonyl of Thr165 and the main chain nitrogen of Y146,

and one between the main chain carbonyl of Arg190 and the main chain nitrogen of Asn167 (Fig. 6B). The stability of the LRR structure is highly dependent on the hydrogen-bond network established by the main chain atoms[54–56], therefore substrate-binding induced disruption of the polar-interaction network might affect the stability of LRR-CT. Mutating Arg166 to alanine (IpaH9.8-R166A) caused a considerable reduction in hGBP1 ubiquitination (Fig. 6C). Moreover, microscale thermophoresis (MST) revealed that IpaH9.8-R166A bound to hGBP1 with a similar dissociation constant ($K_d$) as a wild-type enzyme (Supplementary Fig. 7A), which indicated that, although this

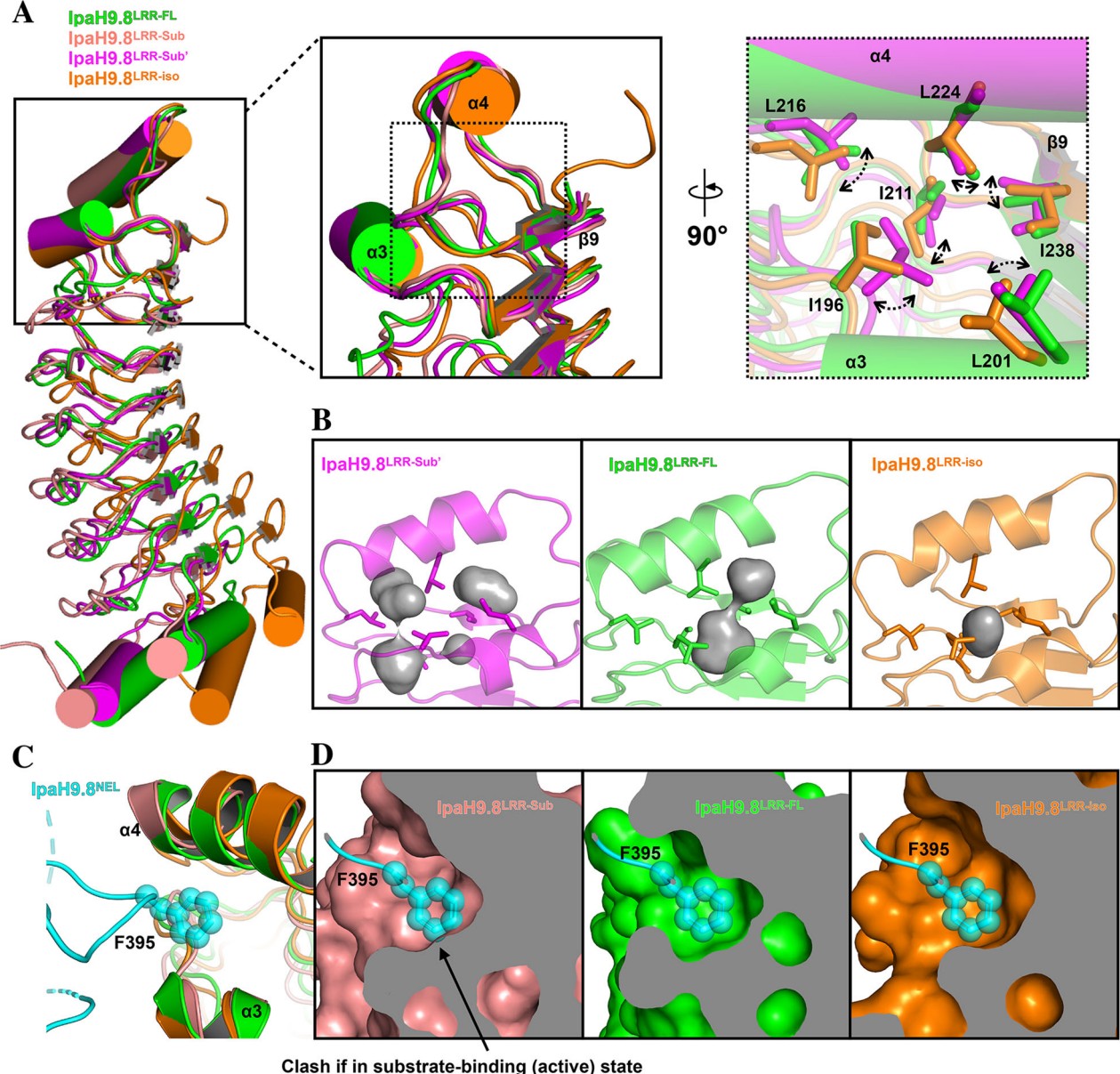

**Fig. 5 Comparison of the hydrophobic pocket of LRR-CT in different IpaH9.8$^{LRR}$ structures. A** Left, superimposition of the structures of IpaH9.8$^{LRR-Sub}$, IpaH9.8$^{LRR-Sub'}$ (PDBID: 6K2D), IpaH9.8$^{LRR-FL}$, and IpaH9.8$^{LRR-iso}$ (PDBID: 5B0T) by LRR-CT with the LRR domains differentiated by colors. Middle, magnified view of LRR-CTs in different IpaH9.8$^{LRR}$ constructs as in left with α3, α4, and β9 labeled. Right, further magnified view of LRR-CTs in different IpaH9.8$^{LRR}$ constructs as in middle but rotated ~90°. The hydrophobic cluster residues were labeled and represented as sticks. Arrows and dash lines indicate the movements of the hydrophobic cluster residues when bound to hGBP1. **B** The central cavities in the LRR-CT of IpaH9.8$^{LRR-Sub'}$ (left), IpaH9.8$^{LRR-FL}$ (middle) and, IpaH9.8$^{LRR-iso}$ (right) were detected using Hollow and a 1.1-Å probe radius[71,72]. The cavities are shown as surface view and colored gray. **C** A close-up view of the LRR–NEL interface between IpaH9.8$^{FL}$ with the LRR-CT of IpaH9.8$^{LRR-Sub}$; IpaH9.8$^{LRR-iso}$ superimposed to IpaH9.8$^{LRR-FL}$. The LRR and NEL of IpaH9.8$^{FL}$ are colored green and cyan, respectively. The IpaH9.8$^{LRR-Sub}$ and IpaH9.8$^{LRR-iso}$ are colored salmon and orange, respectively. The α3 and α4 of LRR domain are labeled. Phe395 of IpaH9.8 $^{FL}$ is labeled and represented as stick and spheres. **D** The hydrophobic pocket of IpaH9.8$^{LRR-FL}$ (left), IpaH9.8 $^{LRR-Sub}$ (middle) and IpaH9.8$^{LRR-iso}$ (right) as in **A**.

substitution reduced ubiquitination, it did not affect the binding affinity of IpaH9.8 for hGBP1. These findings suggested that the Arg166 plays a critical role in mediating the activity of IpaH9.8 by acting as a sensor that reacts to substrate-binding.

Investigation of Phe187 revealed that it may also function as a sensor for detecting substrate-binding in IpaH9.8$^{LRR}$. A magnified view of the LRR-CTs in IpaH9.8$^{LRR-FL}$ and IpaH9.8$^{LRR-iso}$ revealed that the aromatic ring of Phe187, together with the imidazole ring of His210 and the aromatic ring of Tyr239, form a rigorous parallel "π–π–π" stacking interaction, which might be

crucial for the stability of LRR-CT (Fig. 6D, top-left, top-right). Previously, the triple mutant R163A/F187A/H210A was observed to abrogate the toxicity of IpaH9.8 in yeast[31]. It must be noted that, in IpaH9.8$^{LRR-Sub}$, the Phe187 and His210 adopted different conformations that do not have the "π–π–π" stacking interaction (Fig. 6D, bottom-left; Supplementary Fig. 6C, top). Structural analysis revealed that this conformational change was caused by interaction between the Gln110 of hGBP1 with the Phe187 of IpaH9.8$^{LRR}$ (Fig. 6D, bottom-left; Supplementary Fig. 6C, top). Upon this interaction, the side chain Cζ of Phe187 shifted ~3.2 Å

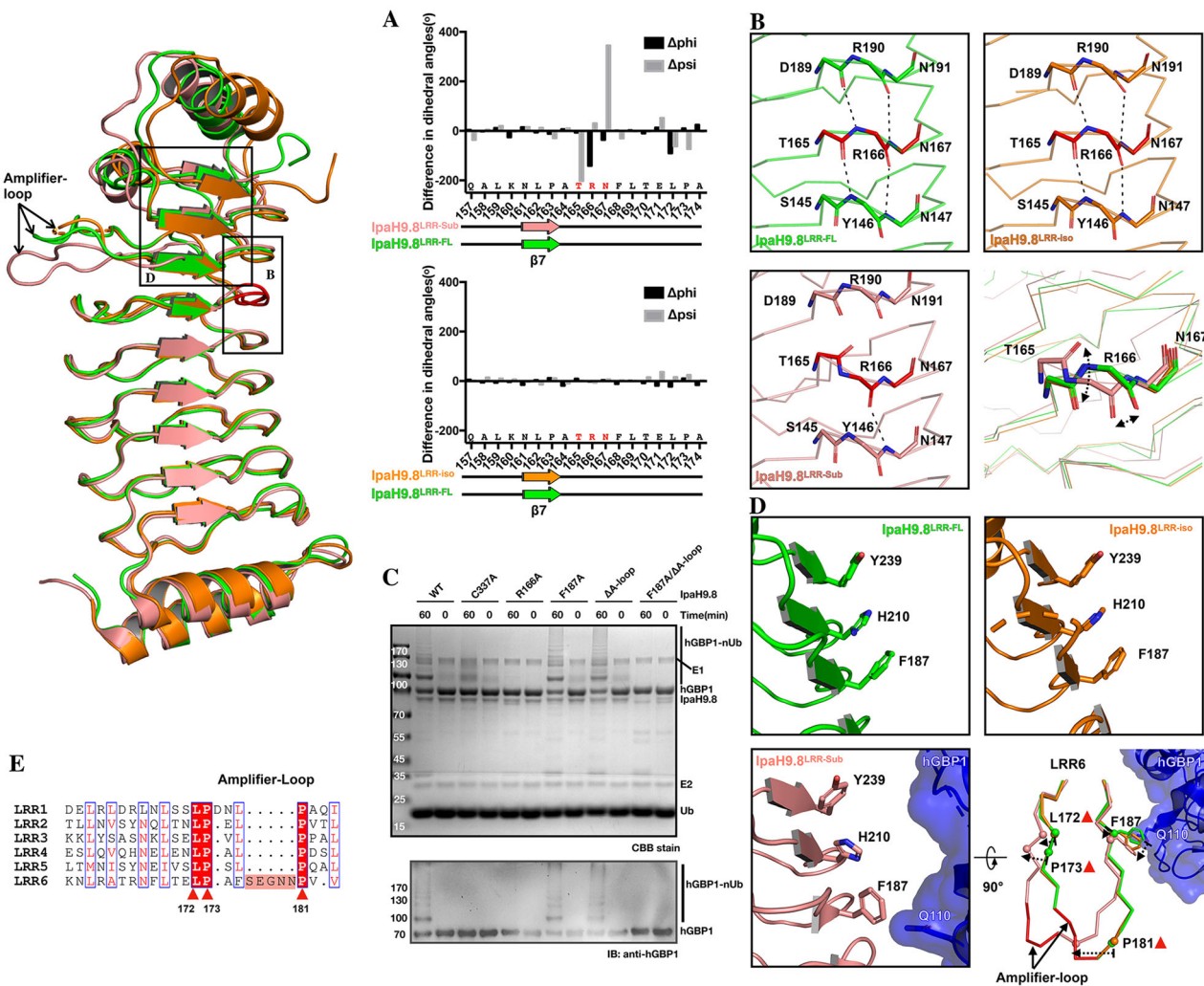

**Fig. 6 Arg166 and Phe187 in IpaH9.8[LRR] are two sensors for substrate-binding. A** Top, differences in phi and psi angles between IpaH9.8[LRR-Sub] and IpaH9.8[LRR-FL] are shown as bar graphs. Bottom, differences in phi and psi angles between IpaH9.8[LRR-iso] (PDBID: 5B0T) and IpaH9.8[LRR-FL] are shown as bar graphs. Cartoons below the graphs indicate secondary structures for IpaH9.8[LRR-Sub] (salmon), IpaH9.8[LRR-iso] (orange), and IpaH9.8[LRR-FL] (green). **B** Magnified view of IpaH9.8[LRR-FL] (green, top-left), IpaH9.8[LRR-iso] (orange, top-right) and IpaH9.8[LRR-Sub] (salmon, bottom-left) near the Arg166 with backbone Cα represented as sticks. Thr165, Arg166, and Asn167 in three structures are colored red. Dashed lines indicated hydrogen bonds. Bottom-right, comparison of the main chain structures of Thr165, Arg166, and Asn167 in different IpaH9.8[LRR] constructs. The main chain of Thr165, Arg166, and Asn167 are represented as sticks. Arrows indicate the main chain movement of Thr165 and Arg166 when bound to substrate. **C** In vitro ubiquitination assays with indicated wild-type or mutants of IpaH9.8. Reaction products were detected by CBB staining (top) and anti-hGBP1 antibody (bottom). **D** Magnified view of IpaH9.8[LRR-FL] (green, top-left), IpaH9.8[LRR-iso] (orange, top-right), and IpaH9.8[LRR-Sub] (salmon, bottom-left) near Phe187, with Phe187, His210, and Tyr239 represented as sticks. Bottom-right, comparison of the LRR6 motifs in different IpaH9.8[LRR] constructs with the backbone Cα shown for LRR6. The Phe187 side chain is represented as sticks. The Ca atom of Leu172, Pro173, Pro181, and Phe187 are shown as spheres. The amplifier-loops (Ser176–Asn180) in IpaH9.8[LRR-Sub] and IpaH9.8[LRR-FL] are indicated and colored red. Arrows indicate the main chain movement of Leu172, Pro173, Pro181, and Phe187, as well as the side chain shift of Phe187 when bound to substrate. hGBP1 is shown as both cartoon and surface (blue, bottom-left and bottom-right, respectively), with Glu110 labeled and represented as sticks. **E** Sequence alignment of the LRR motifs in IpaH9.8. Residues with similar properties are indicated with thin blue transparent boxes and highlighted as red. Strictly conserved residues are shown in red boxes. The amplifier loop is highlighted and in red. Leu172, Pro173, and Pro181 are conserved in LRR motifs and indicated by triangles as in **D** (bottom-right).

and the main chain Cα of Phe187 shifted ~1.3 Å with respect to IpaH9.8[LRR-FL] (Fig. 6D, bottom-right; Supplementary Fig. 6F). Remarkably, following the shift of Phe187, the LRR6 in IpaH9.8[LRR-Sub] exhibited an outward shift toward the convex side with respect to the structures of IpaH9.8[LRR-FL] and IpaH9.8[LRR-iso] (Fig. 6D, bottom-right). In particular, the main chain Cα of LRR-motif conserved residues Leu172, Pro173, and Pro181, which are important for the structure stability of the LRR domain[54,56], translated ~3.1, ~4.1, and ~6.5 Å, respectively, toward the convex side compared with IpaH9.8[LRR-FL] and IpaH9.8[LRR-iso], respectively (Fig. 6D, bottom-right; 6E). In order

to explore why the structural shift of Phe187 in hGBP1 could trigger a dramatic conformational change in the entire LRR6 motif, sequences comparison was performed for the LRR motif. Unlike LRR1-5, LRR6 of IpaH9.8 contains an extra loop (Ser176–Asn180) (Fig. 6E) and structural comparison of the LRR6 motif in three LRR structures revealed that this loop region relieves the structural tension caused by the Phe187 shift, allowing greater flexibility of LRR6 (Fig. 6D, bottom-right). This suggested that this loop functions as an amplifier for conformational changes in the entire LRR6 motif. The remarkable structural rearrangement of LRR6 caused by interaction between

Phe187 and the loop region (named Amplifier-loop, A-loop) might disrupt LRR-CT stability. Indeed, although the mutants IpaH9.8-F187A and IpaH9.8-ΔA-loop had no effect on the hGBP1 ubiquitination activity of IpaH9.8, the double mutant (IpaH9.8-F187A/ΔA-loop) abolished hGBP1 ubiquitination (Fig. 6C). Taken together, these results revealed that both Arg166 and Phe187 may function as sensors for substrate-binding. The substrate-binding destabilizes LRR-CT and consequently activates IpaH9.8 by triggering conformational changes in Arg166 and Phe187. In support of this hypothesis, Arg166 and Phe187 in IpaH9.8$^{LRR}$–hGBP1$^{LG-MD}$ behave similarly as in IpaH9.8$^{LRR}$–hGBP1 (Supplementary Fig. 6B–F), demonstrating that these two residues serve as important sensors.

**The dynamic orientation of LRR relative to NEL is important for Ub delivery from NEL to the substrate.** A previous study probed binding interactions between the E2–Ub intermediate and the SspH1 NEL domain, and proposed a model for the IpaH protein family in which Ub is delivered from E2 to E3 by cooperation between NEL subdomains[57]. However, the molecular mechanism of IpaH proteins for catalyzing the Ub transfer from the NEL cysteine to substrate remains unclear. Superimposition of the structures of IpaH9.8 and other IpaH proteins using the Cα atoms of the N-subdomain and M-subdomain revealed that the C-subdomains from different IpaH proteins spanned over ~45°, demonstrating dynamic behavior in the NEL subdomains (Fig. 7A). Further structural comparison revealed that the LRR domains of IpaH3, Slrp, and SspH2 rotated ~10°, ~15°, and ~160° with respect to IpaH9.8, respectively (Fig. 7A). Even more remarkably, the linker regions between the LRR and NEL in almost all solved IpaH proteins are disordered, except for Slrp due

to binding with the Trx1 substrate, indicating that the linkers are flexible (Figs. 1A and 7B). As LRR plays a role in substrate recognition, the high range rotational movement of LRR may help E3 to transfer Ub to the substrate. To test this hypothesis, IpaH9.8 mutants about the linker region were generated and then subjected to functional analysis. Deletion of the LRR–NEL linker (IpaH9.8$^{ΔLinker}$, deletion of $^{244}$DGQQNTLHRP$^{253}$) completely abolished ubiquitination activity in Ipah9.8 for hGBP1, but did not affect the polymerization of Ub chain (Fig. 7C), suggesting that this linker is critical for Ub transfer from NEL to substrate. Circular dichroism (CD) spectroscopic results indicated no distinct conformational changes between the wild-type and IpaH9.8$^{ΔLinker}$ (Supplementary Fig. 7B). Additionally, substitution of the linker of IpaH9.8 with both (GS)$_n$ repeat (IpaH9.8$^{GS-Linker}$, $^{244}$DGQQNTLHRP$^{253}$ to $^{244}$GGSGSGSGSG$^{253}$) and the linker of SspH1 (IpaH9.8$^{SspH1-Linker}$, $^{244}$DGQQNTLHRP$^{253}$ to $^{244}$GASAPRETRA$^{253}$) recovered the activity of IpaH9.8 for hGBP1 ubiquitination (Fig. 7C). Together, these results provide strong evidence that the dynamic orientation of LRR relative to NEL in IpaH9.8 is important for Ub transfer from the NEL to the substrate.

## Discussion

In this study, we illuminated the key role of the LRR domain in autoinhibition of IpaH9.8 and revealed that IpaH9.8 can be activated by its LRR domain through substrate binding. Structural and biochemical analyses suggest a scenario for this consecutive activation progress: (1) In apo form, the hydrophobic cluster in the LRR-CT of IpaH9.8 forms a hydrophobic pocket to bind NEL and inhibits its catalytic activity (Fig. 8, left); (2) then, the substrate is recruited by the LRR-NT of IpaH9.8 and induces

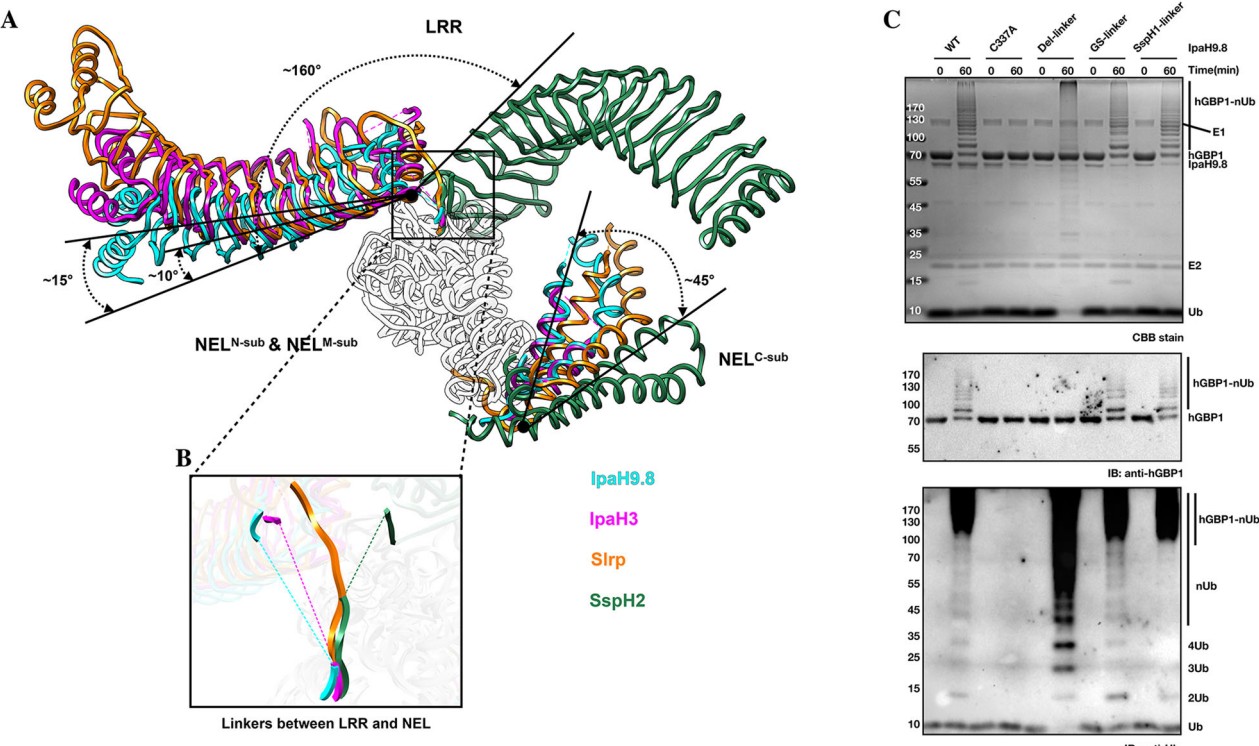

**Fig. 7 The dynamic orientations of LRR relative to NEL in IpaH9.8. A** Superimposition of the structures of IpaH9.8, IpaH3 (PDBID: 3CVR), SspH2 (PDBID: 3G06), and Slrp (PDBID: 4PUF) by N-subdomain and M-subdomain of NEL, with enzymes differentiated by colors. The NEL$^{N-sub/M-sub}$ of IpaH enzymes are colored gray. Arrows indicate the different rotation angles of LRR in different IpaH enzymes. **B** Different linkers between LRR and NEL in different IpaH enzymes. The invisible linkers are represented by dashed lines. **C** In vitro ubiquitination activity assays with wild-type or mutant IpaH9.8. Reaction products were detected by CBB staining (top), anti-hGBP1 antibody (middle), and anti-ubiquitin antibody (bottom).

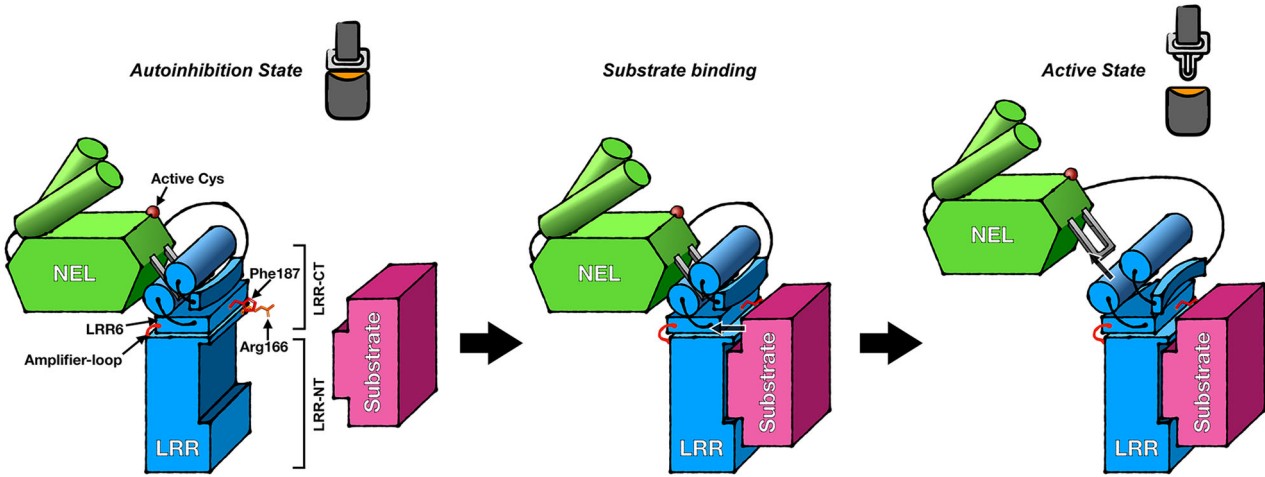

**Fig. 8 Model for substrate-binding-induced activation of IpaH9.8.** (Left) The apo-form IpaH9.8 is autoinhibited (inactive state) because the NEL domain is bound by LRR-CT. (Middle) Substrate binding to LRR-NT of IpaH9.8 triggers the LRR-CT movement by interacting with two sensor residues, i.e. Arg166 (orange) and Phe187 (red). (Right) The conformational change results in destabilization of LRR-CT and dissociation of NEL (active state). The LRR and NEL of IpaH9.8 are colored blue and green, respectively. The substrate is colored pink. The active cystine, Arg166, Phe187, LRR6 motif, and amplifier-loop are indicated by arrows.

conformational changes of two key sensor residues, i.e. Arg166 and Phe187, which subsequently lead to a movement of the LRR6 motif (Fig. 8, middle); (3) the movements of Arg166 and Phe187 trigger a conformational change in LRR-CT and result in the destabilization of the hydrophobic cluster, and, finally, the NEL is released from the hydrophobic cluster, which relieves the auto-inhibition of IpaH9.8 (Fig. 8, right).

IpaH enzymes that are autoinhibited in mode 1 and mode 2 are intrinsically different in both overall structural arrangements, where LRR and NEL are in different relative orientations[27,31]. In our study, we showed that they also have different detailed structural features. For example, the extra loop in LRR6 of IpaH9.8 functions as an amplifier, which is important in substrate-induced activation, and is also conserved in the LRR domain of IpaH3 (Supplementary Fig. 8)[28]. This differs from SspH1 and SspH2, in which each LRR motif is conserved in amino acid sequence and no such loop region exists in their LRR domains (Supplementary Fig. 8)[29,31]. It must be noted that both mode 1 and mode 2 might operate in IpaH enzymes in equilibrium[27], but the observed structural differences in the LRR domains of IpaH3 and SspH1/SspH2 suggest that the activation mechanism of IpaH9.8 might be plausibly applicable to IpaH3. Therefore, whether both the inhibitory modes operate in the same enzyme remains to be elucidated and explored. Future structural work focusing on the IpaH enzymes in the autoinhibition state might give more precise answers.

In an earlier study, Edwards et al. demonstrated that IpaH9.8 shows cooperative kinetics requiring the enzyme to exist as oligomer, inferred the IpaH9.8 oligomerization interface and the corresponding important residue based on the structure of IpaH3[46]. In our Native-PAGE experiment, we confirmed that IpaH9.8 can form oligomers. Our experiments also demonstrated that the oligomerization of IpaH9.8 is irrelevant to the LRR domain (Supplementary Fig. 1A–C). Consistent with the Native-PAGE results, further structural analysis indicated that IpaH9.8 forms a NEL-mediated symmetric dimer or even tetramer in crystals (Supplementary Fig. 9A). However, the oligomerization pattern in the crystal structure of IpaH9.8 is different from that of IpaH3 and IpaH1.4[28,30] (Supplementary Fig. 9B, C). In addition, the residues that were presumed to be important for the oligomerization of IpaH3 were far away from the oligomerization interface in the crystal structures of IpaH9.8 and IpaH1.4

(Supplementary Fig. 9). There are two possible explanations for these results: (1) The oligomers of different IpaHs may form in different ways; and (2) each IpaH protein may have more than one oligomer formation pattern. Determining more IpaH protein structures may provide information that we could use to under-stand the process of IpaH9.8 oligomerization and identify the detailed molecular mechanisms underlying this process.

Previously, IpaH enzymes have been suggested as promising targets for antibacterial drugs or as vaccine candidates due to their critical roles in pathogenesis[22]. The highly conserved NEL domain, in particular, has been identified in multiple IpaH enzymes and could be an ideal target for broad-spectrum anti-biotics[22]. This study, however, revealed that IpaH9.8 could affect the host's antibacterial function by preventing hGBP1 oligomer-ization even in the absence of its catalytic activity, indicating that IpaH9.8 could perform noncatalytic functions through engage-ment of host proteins. Moreover, other IpaH enzymes have been observed previously to engage in noncatalytic functions[31,58–61]. Targeting the LRR domain of the IpaH enzymes may disrupt their substrate-binding ability and could be a better choice for effectively developing live bacterial vaccines and antibacterial drugs.

A previous study proposed targeting the binding region of the E2 enzyme and highlighted the importance of the dynamics of the NEL domain for the Ub transfer[57]. In this study, we further confirmed that the dynamic relative orientations of LRR and NEL are important for Ub transfer from NEL to substrate. More intermediate-state structures will be necessary to understand the detailed molecular mechanism of Ub transfer during the ubi-quitylation catalyzed by IpaH enzymes.

## Methods

**Plasmids**. The wild-type (WT) gene of *S. flexneri* IpaH9.8, the WT gene of *human* Ub, and the codon-optimized gene of *human* GBP1 (hGBP1) were synthesized by GENEWIZ, China. The cDNA of *human* UbcH5b and *mouse* UBA1 were provided by Prof. Feng Rao (Southern University of Science and Technology, China) and Prof. Naixia Zhang (Shanghai Institute of Materia Medica, Chinese Academy of Sciences, China), respectively. IpaH9.8, IpaH9.8$^{\Delta N21}$(21–545), IpaH9.8$^{LRR}$ (21–244), IpaH9.8$^{NEL}$ (254–545), hGBP1, Ub, and UbcH5b were cloned into pET28M, an *Escherichia coli* expression vector with the original N-terminal thrombin protease site replaced by a Tobacco Etch Virus protease site. *Mouse* UBA1 was cloned into pET28a. All mutants were introduced using PCR-based mutagenesis and confirmed by sequencing.

**Protein expression and purification**. Protein constructs for crystallization and in vitro ubiquitination activity assays were expressed in *E. coli* strain BL21 (DE3) Codon Plus (Stratagene). Large-scale cultures were grown in LB medium at 37 °C until the optical density at 600 nm reached ~0.6. Overexpression of proteins was then induced by the addition of isopropyl-β-D-thiogalactopyranoside to a final concentration of 0.25 mM. After growing at 18 °C for 16 h, the cells were harvested by centrifugation at 4500×*g* for 30 min at 4 °C. The cell pellets were washed twice with lysis buffer containing 50 mM Tris-HCl (pH 8.0), 500 mM NaCl, and 0.5 mM dithiothreitol, and then stored at −80 °C until use.

For the purification of the target proteins, the harvested cell pellets were resuspended in lysis buffer that was supplemented with 0.1 mg/mL DNase (Sigma), 0.5 mg/mL lysozyme (Sigma), and a protease inhibitor cocktail (complete ETDA-free, Roche), and disrupted by sonication. The cell lysate was centrifuged at 40,000×*g* for 30 min at 4 °C. The filtered supernatant was then loaded onto a Ni-affinity column (5 mL Histrap™ HP, GE Healthcare) pre-equilibrated with lysis buffer using an AKTA pure system (GE Healthcare). The sample was eluted with a gradient of 0–500 mM imidazole in lysis buffer. Proteins for crystallization were further purified by anion exchange using a 5-mL Histrap™ Q column (GE Healthcare), and subsequently loaded onto a HiLoad 16/60 Superdex 200 (GE Healthcare) pre-equilibrated with buffer containing 50 mM Tris-HCl (pH 7.5), 100 mM NaCl, and 1 mM dithiothreitol. Proteins used for in vitro ubiquitination activity assays were directly subjected to gel filtration by Superdex 200 or Superdex 75 columns. The peak fractions were concentrated using an Amicon Ultra-15 column, snap frozen in liquid nitrogen, and stored at −80 °C. All purification steps were performed at 4 °C and the eluted proteins were examined using SDS-PAGE.

**Crystallization and data collection**. IpaH9.8 (22-537) was concentrated to 25 mg/mL and the initial crystallization screening was performed using a sitting-drop vapor-diffusion method with a series of crystallization kits from Molecular Dimensions (Newmarket, England) in 96-well plates. Drops were prepared by mixing 0.75 μL of both protein solution and reservoir solution together, followed by equilibration with 75 μL of reservoir solution at 20 °C. Appropriate crystals were obtained within 1 week at 20 °C using a reservoir solution containing 2.5 M sodium chloride and 0.1 M potassium/sodium phosphate buffer (pH 6.0).

Prior to applying the crystallization screens for the IpaH9.8$^{LRR}$ (22–244) and hGBP1 (full-length) complex, 0.3 mM IpaH9.8$^{LRR}$ was mixed with 0.3 mM hGBP1 and 5 mM GDP. Sitting drops screens were set as above. The first crystals could be observed after 3 days. After optimizing the conditions, appropriate crystals were obtained within 1 week at 20 °C using a reservoir solution containing 3.5 M sodium chloride and 0.1 M sodium citrate buffer (pH 6.5).

All crystals were soaked in a cryoprotectant solution containing 20% (v/v) ethylene glycol and then snap frozen in liquid nitrogen prior to shipment. Diffraction data from the crystals of IpaH9.8$^{Δ21}$ and the IpaH9.8$^{LRR}$ and hGBP1 complex were collected using the beamline BL17U at SSRF (Shanghai, China) at 0.97 Å wavelength and 100 K temperature. All data sets were indexed, integrated, and scaled using the XDS program suite[62]. All data collection statistics are summarized in Table 1.

**Structure solution and refinement**. The structure of IpaH9.8 was determined using the molecular replacement method with the program *Phaser* in the *Phenix* program package[63]. The structure of IpaH9.8$^{LRR}$ (PDB: 5B0T) was used as the search model. Partial density was visible for IpaH9.8$^{NEL}$. *Sculptor* in the *Phenix* program suite was used to create a model of IpaH9.8$^{NEL}$ using IpaH3$^{NEL}$ (PDB: 3CVR) as a base structure, which was placed into the available density. Several rounds of refinement were performed using the *phenix.refine* program in the *Phenix* program suite, alternating with manual fitting and rebuilding based on $2F_o − F_c$ and $F_o − F_c$ electron density using *COOT*[63,64].

The structure of the IpaH9.8$^{LRR}$ and hGBP1 complex was determined using the structure of IpaH9.8$^{LRR}$ (PDB: 5B0T) and hGBP1 (PDB: 1DG3) in a molecular replacement method[63]. Subsequently, the GDP molecular structure was constructed using the $2F_o − F_c$ and $F_o − F_c$ electron densities. The model was refined via iterative rounds of refinement and rebuilding using *Phenix*, *Refmac5*, and *COOT*[63–65].

Two crystal structures determined in this study display Ramachandran statistics with 91.59% and 91.46% of residues in the most favored regions and 8.19% and 8.15% of residues in the allowed regions of the Ramachandran diagram, respectively. The final refinement statistics and geometry of both crystal structures defined by MolProbity[66] are shown in Table 1. The molecular graphics representations were generated using *Pymol* or *Chimera*[67,68]. The electron density maps were constructed in *Pymol* from $2F_o − F_c$ maps contoured at the sigma level as indicated.

**In vitro ubiquitination activity assays**. Ubiquitination assays for substrate were performed in PBS buffer (pH 7.4) at 25 °C using 0.25 μM E1 (mouse UBA1), 2 μM E2 (UbcH5b), 0.5 μM E3, 50 μM Ub, 5 mM MgCl₂, 5 mM ATP, and 2 μM hGBP1. Reactions were initiated by the addition of ATP. Sampled time points were taken by adding 20 μL of the reaction mixture to 20 μL of reducing SDS-PAGE load dye, followed by boiling prior to electrophoresis. SDS-polyacrylamide gels were loaded with 20 μL of sample, followed by electrophoresis, Coomassie staining, and imaging

## Table 1 Data collection and refinement statistics (molecular replacement).

|  | IpaH9.8 | IpaH9.8$^{LRR}$–hGBP1 |
|---|---|---|
| Number of xtals | 1 | 3 |
| Data collection |  |  |
| Space group | *I222* | *P6₁22* |
| Cell dimensions |  |  |
| *a, b, c* (Å) | 106.64, 122.25, 149.97 | 112.49, 122.49, 582.35 |
| *α, β, γ* (°) | 90, 90, 90 | 90, 90, 120 |
| Resolution (Å) | 2.75 (2.85–2.75)* | 3.72 (3.85–3.72) |
| $R_{sym}$ or $R_{merge}$ | 0.080 (1.184) | 0.3248 (4.069) |
| $I / σI$ | 15.74 (2.56) | 12.52 (2.59) |
| Completeness (%) | 99.46 (99.41) | 99.65 (99.38) |
| Redundancy | 6.6 (7.0) | 37.3 (37.5) |
| Refinement |  |  |
| Resolution (Å) | 86.91–2.75 | 60.91–3.72 |
| No. reflections | 25703 (2513) | 28371 |
| $R_{work} / R_{free}$ | 24.61/28.49 | 22.73/25.56 |
| No. atoms |  |  |
| Protein | 3460 | 6101 |
| Ligand/ion | 0 | 28 |
| Water |  |  |
| B-factors |  |  |
| Protein | 102.76 | 192.19 |
| Ligand/ion |  | 181.13 |
| Water |  |  |
| R.m.s. deviations |  |  |
| Bond lengths (Å) | 0.004 | 0.004 |
| Bond angles (°) | 1.09 | 1.00 |

*Values in parentheses are for highest-resolution shell.

on a Bio-Rad Gel Doc (Bio-Rad). For immunoblot analysis, SDS-polyacrylamide gels were loaded with 5 μL of sample, followed by electrophoresis, transfer to a PVDF membrane, and incubated with specific primary antibodies as indicated. Ubiquitination assays for the polymerization of free Ub chains were performed as above but without hGBP1 and detected by Coomassie staining. Finally, the signals were visualized by the chemiluminescence system (Perkin Elmer). These experiments were conducted three times. The uncropped and unprocessed western blotting images were show in Supplementary Fig. 11.

**B-factor normalization**. The normalized *B* factor of Cα atoms were calculated by Eq. (1) to estimate the structure dynamics for each IpaH9.8$^{LRR}$ in this study[69,70].

$$B^i_{norm} = \frac{B^i − \bar{B}}{δ_B} × \frac{1}{1.645},\tag{1}$$

$$\ddot{B}^i_{norm} = \min\left[\max\left(B^i_{norm}, −1\right), 1\right]\tag{2}$$

where $B^i$ is the B factor of Cα atom *i*, $\bar{B}$ and $δ_B$ are the mean and the standard deviation of the B factor, respectively, of all atoms within a binding unit of the PDB biological complexes, and $B^i_{norm}$ is the normalized B factor of atom *i*. The number 1.645 is a typical threshold under a standard normal distribution, indicating the 0.05 probability of a value outside of [−1.645, 1.645] for each of the two tails; *min* means the minimum of two values, while *max* returns the maximum. Equation (1) was used to normalize and scale the 90% confidence interval of the B factor to [−1, 1]. Equation (2) was used to set any value outside the 90% confidence interval to either −1 or 1, whichever was closer.

**Microscale thermophoresis**. MST binding assays were performed using 75 nM NT-647-NHS-labeled hGBP1 diluted in PBS buffer (pH 7.4). Purified IpaH9.8-WT and IpaH9.8–R166A were titrated into fluorescently labeled hGBP1 and incubated for 30 min at room temperature before MST scans were performed using a Monolith NT. Automated device (NanoTemper Technologies). Binding curves were fit using NT Analysis Software. The results revealed that IpaH9.8 mutant R166A displayed similar curves to the wild-type IpaH9.8 (Supplementary Fig. 8A), suggesting that the mutant had similar binding affinity to hGBP1 with the wild-type enzyme.

**CD spectroscopy**. CD spectroscopy was performed to determine whether the differences in catalytic activity between mutants and wild-type enzymes were due to protein unfolding. CD spectra were collected using a J800 spectropolarimeter (Ja-156 pan, Spectroscopic Company) under a N₂ atmosphere at room temperature

in a quartz cell with a 1-mm path length. The purified samples (IpaH9.8-WT, IpaH9.8-ΔLinker, IpaH9.8–R166A, IpaH9.8-F187A, and IpaH9.8-F187A/ΔA-loop) were dialyzed in 10 mM sodium phosphate buffer at pH 7.5 and at 4 °C overnight before measurement. The CD spectra were obtained from a wavelength region of 190–260 nm, taking the average of four scans. The molar ellipticity per residue was calculated by $(h) = h / (10n \times c \times l)$. Here, $h$ is the CD signal in mdeg, $n$ the number of residues, $c$ the concentration in mol/L, and $l$ the length of the cuvette path (cm). The sample concentrations were measured using a Quant-iTTM Protein Kit (Invitrogen) three times, and the average values were applied to calculate the molar ellipticity per residue. The result showed that all IpaH9.8 mutants display the same curves with the wild-type IpaH9.8 (Supplementary Fig. 8B), suggesting that these mutants should be folded similar to wild-type enzymes in solution.

**Statistics and reproducibility**. Each experiment for ubiquitination assay was repeated three times, and sample sizes and numbers are indicated in detail in the Methods section.

**Reporting summary**. Further information on research design is available in the Nature Research Reporting Summary linked to this article.

## Data availability

The coordinates and structure factors for IpaH9.8 and IpaH9.8$^{LRR}$-hGBP1 have been deposited in the RCSB Protein Data Bank (PDB) with the accession number 6LOL and 6LOJ, respectively. Source data underlying plots shown in Fig. 6 are provided in Supplementary Data 1. All relevant data are available upon request.

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

## Acknowledgements

We thank Dr. Huan Zhou of the beamline BL-17U at Shanghai Synchrotron Radiation Facility (SSRF) for assistance with data collection. We thank Prof. Feng Rao and Prof. Naixia Zhang for providing plasmids. We thank Ziyang Fu for helpful discussions. This work was funded by the National Natural Science Foundation of China (21907006), the Natural Science Foundation of Guangdong Province (2020A1515011544), the Shenzhen Science and Technology Innovation Committee (JCYJ20180302150357309) to Y.Y., and the Shenzhen Science and Technology Innovation Committee (JCYJ20170412150913708) to H.H. We thank LetPub (www.letpub.com) for its linguistic assistance during the preparation of this manuscript.

## Author contributions

Y.Y. and X.Y. performed the biochemical experiments. Y.Y. contributed to crystal-lographic calculations. Y.Y. and H.H. designed the study and planned the experiments. Y.Y. and H.H. wrote the paper. All authors discussed the results and commented on the manuscript.

## Competing interests

The authors declare no competing interests.
