## [Peer Review File · Communications Biology]

Reviewers' comments:

Reviewer #1 (Remarks to the Author):

Shigella harbors a family of ten ubiquitin ligases that serve as virulence factors in remodeling host cell proteomes for pathogen survival and replication. These IpaH ligases exhibit marked sequence conservation that manifests as structural conservation, as one would expect. The most complete full length structure is that of IpaH3, which serves as a model for the other family members. All members of the IpaH family share a common domain architecture comprised of an N-terminal leucine-rich repeat domain required for binding target protein substrates for appending of polyubiquitin degradation signals assembled by a C-terminal catalytic domain. The structure for the catalytic domain of the closely related IpaH9.8 supports this prediction with respect to that of IpaH3. The catalytic domain contains an absolutely conserved active site cysteine that forms an obligate thioester to ubiquitin as an intermediate in the catalytic cycle, prompting suggestions for similarity with the Hect family of eucaryotic ubiquitin ligases although the two enzyme families show no structural conservation. Until recently, our understanding of the biology of these bacterial ubiquitin ligases exceeded our understanding of the mechanism by which the enzymes function, which is assumed to be conserved among the family members.

In the present manuscript, the authors report the structure of the nearly full length IpaH9.8 and confirm conservation with the prior structure for IpaH3. The authors also extend earlier work by Li et al. (citation 50) on the structure of the human guanylate-binding protein 1 substrate bound to the IpaH9.8 targeting domain and explore the structural differences between the "autoinhibited" and "active states" of IpaH9.8 for free and substrate bound ligases. Unfortunately, although the authors reference the work in an otherwise irrelevant citation in the introduction of the manuscript, the authors appear unaware of the content of an earlier kinetic analysis of IpaH9.8 by Edwards et al. from 2014 (citation 46) that shows the paralog actually to have robust activity in the assembly of free chains that was previously missed because the unanchored chains migrate in the stacker gel while autoubiquitination products of chains linked to the ligase migrate at the very top of the resolving gel during SDS-PAGE. Because the stacker gel is discarded, this activity was missed by previous investigators. Edwards et al go on to show IpaH9.8 shares a two-site mechanism for polyubiquitin chain formation with the Hect ligases termed Proximal Indexation. A prominent feature of the model is the assembly of the polyubiquitin degradation signal on the active site cysteine prior to en bloc transfer to the target protein, in contrast to the assumed mechanism of sequential distal chain assembly. Edwards et al. also show that IpaH9.8 shows cooperative kinetics that is mediated by interdomain contacts mediated through Phe395 (identified by the authors in the present manuscript), requiring the enzyme to exist as an oligomer, and identify a dimeric symmetry pair across unit cells consistent with the cooperative kinetics. [Interestingly, remarks from the Protein Data Bank noted this extensive subunit interface had been identified by a PISA analysis but not noted by Zhu et al. in the original IpaH3 structure paper (citation 28).] The authors of the present manuscript make no note of whether they observe IpaH9.8 as a dimer or how this affects the interpretation of their results. The apparent "activation" of IpaH9.8 that is observed on truncation of the catalytic domain is shown by Edwards et al. to be conversion from the cooperative to a hyperbolic mechanism with loss of the targeting domain. Thus, the present authors could greatly strengthen their manuscript by interpreting their structural data within the context of this extensive earlier mechanistic work. Otherwise, the present observations only make incremental advances on what is already known or reasonably extrapolated from the existing literature.

Reviewer #2 (Remarks to the Author):

The IpaH family proteins from Gram-negative bacteria are unique E3 ubiquitin ligases that target important proteins in the host cell. IpaH9.8 from *Shigella flexneri* ubiquitinates and degrades a subset of GBPs in human cells to suppress host defense. This manuscript by Ye and colleagues

investigated the autoinhibition mechanism of IpaH9.8, as well as its activation induced by substrate-binding, by reporting the crystal structure of full-length IpaH9.8 in the auto-inhibited state, and the crystal structure of IpaH9.8LRR in complex with GBP1. They have identified a hydrophobic contact region between the LRR and NEL domains, which is responsible for the autoinhibitory effect. Their structure of IpaH9.8LRR-GBP1 largely resembles a previously determined IpaH9.8LRR-GBP1LG-MD structure, which consolidates the substrate recognition mechanism of IpaH9.8. Finally, the authors also investigated the function of the linker region between the LRR and NEL domains, which is important for the dynamics and Ub transfer function of IpaH9.8. Overall, the authors did a good job of systematically investigating the structure and function of IpaH9.8. Although the results reported here are not fundamentally novel, they are of some interest and deserve to be published.

A few comments:

1. The authors propose that the previously determined IpaH9.8NEL-iso structure is not in a native state, since it would clash with LRR domain. This is likely to be case, but how do the authors explain the existence of a dimeric IpaH9.8 in non-reducing conditions as reported in reference 48?
2. Figure 2B: The mutation of F395 to R should greatly diminish the autoinhibitory effect of IpaH9.8. How come the activity of this mutant is so weak, when compared to other mutants such as I211D?
3. Table S1: The resolution cutoffs are a bit too aggressive, especially for the complex crystal. I would use an I/sigma 2.0 cutoff, which is the golden standard in crystallography.
4. The authors should also be less careless and consolidate the writing throughout. For example, Figure S2: Label mistakes—should be the NEL domains instead of LRR domains; Page 11, 2nd paragraph in Crystallization and Data Collection, what is RaMP2?

Reviewer #3 (Remarks to the Author):

The IpaH enzymes are a class of bacterially encoded E3 ubiquitin ligases that share no sequence or structural similarity with eukaryotic E3s. These enzymes contribute to the virulence of many important pathogens including Shigella and Salmonella. IpaHs have the ability to catalyze ubiquitin chain formation and also to transfer ubiquitin to eukaryotic substrates of the cells that are infected by the pathogens. In the absence of their substrates the IpaH activity is inhibited. This inhibition is due to the interaction of the LRR domain (N-terminus) with the catalytic NEL domain. There are two modes of this inhibition that have been identified. It has been proposed that one mode of inhibition may be relieved by simple displacement by the substrate. How the other mode of inhibition may be relieved has been an open question and is the focus of this manuscript.

The authors solve the structure of IpaH9.8 in complex with its substrate GBP1. The authors conduct exhaustive mutational analysis to demonstrate the importance of a hydrophobic cluster located in the LRR. They propose a model whereby Arg166 and Phe187 function as "sensors" of substrate binding which then destabilize LRR_NEL interactions and allow for activation of the enzyme. Analysis of IpaH9.8 captured in a variety of different states reveals a dynamic movement of the LRR domains.

The data are of high quality and the interpretations are sound. This study represents the first analysis of how the second mode of inhibition is relieved. It awaits the identification of more IpaH substrates (and their structural elucidation) to reveal if these mechanisms are truly general.

Minor criticisms.

In the introduction (paragraph 4) it should be stated that the Ste7 MAPKK substrate is in yeast.

What was done with NEMO? It is mentioned in the Materials and Methods but nowhere else in the text?

The name of the last section "Dynamic structural features of IpaH9.8 that are involved in Ub transfer" is somewhat misleading. The authors show that the linker domain is required for Ub transfer to substrates, presumably because the LRR has to move around a lot. As written, it could be interpreted that the linker region is physically interacting with Ub, playing a more involved catalytic function.

According to the comments of reviewers, we have revised our manuscript in next points, and highlighted the changes of manuscript:

For the comments of Reviewer #1

1. The authors of the present manuscript make no note of whether they observe IpaH9.8 as a dimer or how this affects the interpretation of their results. The apparent “activation” of IpaH9.8 that is observed on truncation of the catalytic domain is shown by Edwards et al. to be conversion from the cooperative to a hyperbolic mechanism with loss of the targeting domain. Thus, the present authors could greatly strengthen their manuscript by interpreting their structural data within the context of this extensive earlier mechanistic work. Otherwise, the present observations only make incremental advances on what is already known or reasonably extrapolated from the existing literature.

>> We would like to thank the reviewer #1 for his valuable suggestion. As reviewer #1 pointed out, the earlier study demonstrated that IpaH9.8 shows cooperative kinetics requiring the enzyme to exist as a dimer. We performed a series of supplementary Native-PAGE experiments, which confirmed that IpaH9.8 does not only form dimers, but also form higher-ordered oligomers. Further, we demonstrated that the oligomerization of IpaH9.8 is NEL-dependent, but independent on the formation of disulfide bonds. Next, we analyzed various crystal structures of different IpaHs in details, and observed that the oligomerization patterns of different IpaHs in their crystal structures are different. Further structural studies of other IpaHs may provide more information that we may use to understand the details of the underlying molecular mechanism. Relative content was added to the results and discussion sections (P4L5, P9L34, Fig. S1, and Fig. S9).

For the comments of Reviewer #2

Overall, the authors did a good job of systematically investigating the structure and function of IpaH9.8. Although the results reported here are not fundamentally novel, they are of some interest and deserve to be published.

>> We would like to thank reviewer #2 for these comments.

1. The authors propose that the previously determined IpaH9.8NEL-iso structure is not in a native state, since it would clash with LRR domain. This is likely to be case, but how

do the authors explain the existence of a dimeric IpaH9.8 in non-reducing conditions as reported in reference 48?

>> We would like to thank reviewer #2 for the in-depth question. In reference 48, the authors demonstrated that the full-length IpaH9.8 will form dimers under non-reducing conditions. Edwards et al. also mentioned that IpaH9.8 shows cooperative kinetics requiring the enzyme to exist as oligomers (reference 46). In order to explore this point, we performed a series of Native-PAGE experiments. Our experiment showed that IpaH9.8 could form oligomers under both reducing and non-reducing conditions, illustrating that the oligomerization of IpaH9.8 does not rely on the disulfide bonds. Our results also suggested that the oligomerization of IpaH9.8 is NEL-dependent. Related content was added to results and discussions sections (P4L5, P9L34, and Fig. S1).

2. Figure 2B: The mutation of F395 to R should greatly diminish the autoinhibitory effect of IpaH9.8. How come the activity of this mutant is so weak, when compared to other mutants such as I211D?

>> We would like to thank reviewer #2 for these in-depth questions. In the structure of IpaH9.8, Phe395 is surrounded by the hydrophobic cluster on LRR-CT and forms hydrophobic interactions with the residues in this cluster. Mutation of Phe to Arg may not be enough to completely disrupt the hydrophobic interactions between Phe395 and these hydrophobic residues because the aliphatic portion of Arg may still engage in hydrophobic interactions with them. To this end, we mutated Phe395 to a more hydrophilic residue Aspartate (F395D). The results show that F395D completely abolished the autoinhibition of IpaH9.8, and its effect is even more obvious than that of I238D. This result further highlights the importance of the hydrophobic interactions between the hydrophobic cluster and Phe395 for IpaH9.8 autoinhibition. Related results and discussions were added to the Supplementary Information (P2L1 and Fig. S10).

3. Table S1: The resolution cutoffs are a bit too aggressive, especially for the complex crystal. I would use an I/σ 2.0 cutoff, which is the golden standard in crystallography.

>> We have adjusted the resolution cutoffs of our structures to the golden standard, re-submitted the new structures to PDB, and updated the Table S1 based on the new structures. We also revised the text according to the resolutions of new structures (P3L32, P5L15).

4. The authors should also be less careless and consolidate the writing throughout. For example, Figure S2: Label mistakes—should be the NEL domains instead of LRR domains; Page 11, 2nd paragraph in Crystallization and Data Collection, what is RaMP2?

>> We apologize for these mistakes. We have corrected “LRR” to “NEL” in Figure S2. We also revised the text in the Crystallization and Data Collection section accordingly. The text now reads “After optimizing the conditions, appropriate crystals were obtained within one week at 20°C using a reservoir solution containing 3.5 M sodium chloride and 0.1 M sodium citrate buffer (pH 6.5)” (P11L36).

For the comments of Reviewer #3

The data are of high quality and the interpretations are sound. This study represents the first analysis of how the second mode of inhibition is relieved.

>> We would like to thank reviewer #3 for these supportive comments.

1. In the introduction (paragraph 4) it should be stated that the Ste7 MAPKK substrate is in yeast.

>> We have revised the text accordingly. The text now reads “IpaH9.8 targets Ste7 for degradation in order to hijack the host UPS and subsequently inhibit mitogen-activated protein kinase (MAPK)-dependent signaling pathways in yeast²³” (P2L38).

2. What was done with NEMO? It is mentioned in the Materials and Methods but nowhere else in the text?

>> We apologize for the misleading sentence in the Materials and Methods. We have deleted the text (P10L32).

3. The name of the last section "Dynamic structural features of IpaH9.8 that are involved in Ub transfer" is somewhat misleading. The authors show that the linker domain is required for Ub transfer to substrates, presumably because the LRR has to move around

a lot. As written, it could be interpreted that the linker region is physically interacting with Ub, playing a more involved catalytic function.

>> Following this comment, we have changed the title of the last section to “*The dynamic orientation of LRR relative to NEL is important for Ub delivery from NEL to the substrate*” (P8L26). Accordingly, the text in the last sentence of this section now reads “Together, these results provide strong evidence that the dynamic orientation of LRR relative to NEL in IpaH9.8 is important for Ub transfer from the NEL to the substrate” (P9L8). And the Figure legend of Figure 7 has also been changed to “*The dynamic orientations of LRR relative to NEL in IpaH9.8*” (P22L19).

REVIEWERS' COMMENTS:

Reviewer #1 (Remarks to the Author):

The authors have addressed my earlier concerns.

Reviewer #2 (Remarks to the Author):

The authors have addressed all my previous concerns.

Reviewer #3 (Remarks to the Author):

The authors have sufficiently addressed my concerns.

Author Responses

For the comments of Reviewer #1

The authors have addressed my earlier concerns.

>> Thank you for your technical review.

For the comments of Reviewer #2

The authors have addressed all my previous concerns.

>> Thank you for your technical very careful review and comments.

For the comments of Reviewer #3

The authors have sufficiently addressed my concerns.

>> Thank you for your very careful review and suggestions.